# Exploring All-In-One Knowledge Distillation Framework for Neural Machine Translation

**Zhongjian Miao[1*]   Wen Zhang[2]   Jinsong Su[1†]   Xiang Li[2]**
**Jian Luan[2]   Yidong Chen[1†]   Bin Wang[2]   Min Zhang[3]**
[1]School of Informatics, Xiamen University, China
[2]Xiaomi AI Lab, China
[3]Institute of Computer Science and Technology, Soochow University, China
miaozhongjian@stu.xmu.edu.cn, {jssu,ydchen}@xmu.edu.cn

## Abstract

Conventional knowledge distillation (KD) approaches are commonly employed to compress neural machine translation (NMT) models. However, they only obtain one lightweight student each time. Consequently, we have to conduct KD multiple times when different students are required at the same time, which could be resource-intensive. Additionally, these students are individually optimized, and thus lack interactions with each other, leading to their potential not being fully exerted. In this work, we propose a novel All-In-One Knowledge Distillation (AIO-KD) framework for NMT, which generates multiple satisfactory students at once. Under AIO-KD, we first randomly extract fewer-layer subnetworks from the teacher as the sample students. Then, we jointly optimize the teacher and these students, where the students simultaneously learn the knowledge from the teacher and interact with other students via mutual learning. When utilized, we re-extract the candidate students, satisfying the specifications of various devices. Particularly, we adopt carefully-designed strategies for AIO-KD: 1) we dynamically detach gradients to prevent poorly-performed students from negatively affecting the teacher during the knowledge transfer, which could subsequently impact other students; 2) we design a two-stage mutual learning strategy, which alleviates the negative impacts of poorly-performed students on the early-stage student interactions. Extensive experiments and in-depth analyses on three benchmarks demonstrate the effectiveness and eco-friendliness of AIO-KD. Our source code is available at https://github.com/DeepLearnXMU/AIO-KD.

## 1 Introduction

In recent years, Transformer (Vaswani et al., 2017) has become the dominant architecture in neural machine translation (NMT). To further improve model performance, there have been many efforts in exploring wider or deeper Transformers (Wang et al., 2019; Xu et al., 2021; Wang et al., 2022). However, deploying such models with billions of parameters on edge devices (e.g., mobile phones) remains to be a challenge. To solve this problem, various methods for model compression have been proposed (Choudhary et al., 2020). Among them, knowledge distillation (KD) (Hinton et al., 2015) has been widely adopted due to its effectiveness and simplicity. In this regard, researchers explore many KD approaches for NMT, such as Word-KD (Kim and Rush, 2016) and Selective-KD (Wang et al., 2021).

Generally, in these conventional KD approaches, the knowledge of a large teacher model is transferred to only a compact student model, which usually adopts the same model architecture as the teacher but with fewer parameters (Kim and Rush, 2016; Jafari et al., 2021; Wang et al., 2021; Yang et al., 2022). Nevertheless, due to hardware differences, we are often required to deploy models of varying sizes on different devices (Sandler et al., 2018; Wang et al., 2020a; Tan et al., 2022). In this scenario, conventional KD approaches have two defects: 1) they have to be conducted multiple times for different students, which leads to substantial costs; 2) students are individually optimized, and thus they are unable to interact with each other. Note that in the human learning process, communication between students will benefit their learning (Webb, 1989). In light of this, we believe that these students can be further improved through collaborative interactions.

In this paper, we propose a novel All-In-One Knowledge Distillation (AIO-KD) framework for NMT, which constructs students from the teacher and jointly optimizes both the teacher and students from scratch. Employing AIO-KD, we regard the fewer-layer subnetworks extracted from the teacher as the candidate students. During training, we first

---

*Work was done when interning at Xiaomi AI Lab.
†Corresponding Author.

randomly select from all candidate students to obtain the sample ones at each training step. Then, we jointly optimize the teacher and the sample students. During this process, the students simultaneously learn the knowledge from the teacher and interact with other students via mutual learning. When utilized, we re-extract the candidate students from the teacher, satisfying the specifications of various devices.

To better accommodate AIO-KD, we carefully design the following strategies:

1) **Dynamic Gradient Detaching.** Under AIO-KD, the students are optimized jointly with the teacher, where the teacher and students mutually influence each other through the KD loss. When there exists a significant performance gap between a student and the teacher, the gradients of the KD loss specific to the student will harm the teacher, which further negatively affects other students. To address this issue, we measure the performance gap through the cross-entropy ratio of the student to the teacher. If this ratio exceeds a pre-defined threshold, we will not utilize these gradients to update the teacher's parameters.

2) **Two-Stage Mutual Learning.** As mentioned previously, we introduce mutual learning to facilitate students. To avoid the negative impacts of poorly-performed students on student interactions at the early stage, we adopt a two-stage training strategy. Concretely, we first only utilize the signals from the teacher to guide the training of students, and further introduce mutual learning to strengthen the interactions between students. Such multi-stage training strategy has been verified in previous study (Zhou et al., 2022a).

Empirical experiments and in-depth analyses on three translation benchmarks demonstrate that AIO-KD is superior to conventional KD approaches in terms of translation quality and training costs. As a bonus, the teacher in AIO-KD is significantly enhanced through knowledge transfer with the students.

## 2 Related Work

Our related works mainly include the following three lines of studies:

**Transformer with Variable Depths.** To reduce computation costs, plenty of researchers investigate the variable-depth Transformer architecture (Yu et al., 2019; Dehghani et al., 2019; Hou et al., 2020; Xin et al., 2020; Liu et al., 2020a; Fan et al., 2020; Elbayad et al., 2020; Cai et al., 2020; Liu et al.,

2021b). However, the majority of these works focus on the Transformer encoder, while paying less attention to the overall Transformer architecture. Our work focuses on the latter, i.e., the overall Transformer with variable depths for NMT.

**Knowledge Distillation in NMT.** In recent years, model compression technologies have attracted much attention (Han et al., 2016; See et al., 2016; Choudhary et al., 2020). As a commonly-used technology for model compression, knowledge distillation (KD) (Hinton et al., 2015) has been widely used in many natural language processing tasks (Jiao et al., 2020; Wang et al., 2020b; Liu et al., 2020b; Zhou et al., 2022c; Zhang et al., 2023). In the community of NMT, Kim and Rush (2016) first apply KD to autoregressive NMT. Further, many studies explore more effective KD approaches for NMT (Zeng et al., 2019; Wei et al., 2019; Zhang et al., 2019; Wang et al., 2021; Liang et al., 2022; Miao et al., 2022; Zhou et al., 2022b). Meanwhile, researchers focus on applying KD to various aspects of NMT, including multilingual NMT (Tan et al., 2019; Do and Lee, 2022; Lu et al., 2022; Huang et al., 2022b, 2023), unsupervised NMT (Sun et al., 2020; Nguyen et al., 2021), non-autoregressive NMT (Gu et al., 2018; Qian et al., 2021; Huang et al., 2022a; Wang et al., 2023), and $k$NN-NMT (Yang et al., 2022).

**Mutual Learning in NMT.** Mutual learning (Zhang et al., 2018) has been explored in NMT, with various techniques proposed to improve translation quality. For example, Bi et al. (2019) propose multi-agent learning, where diverse students learn from each other, working together to improve translation quality. Liao et al. (2020) explore sentence-level and token-level mutual learning for NMT. Zhao et al. (2021) show the effectiveness of mutual learning in end-to-end speech translation.

To the best of our knowledge, our work is the first attempt to incorporate both knowledge distillation and mutual learning into the variable-depth Transformer. Unlike conventional KD approaches, the candidate students in AIO-KD are the fewer-layer subnetworks derived from the teacher. During training, we randomly select from all candidate students to obtain the sample ones and jointly optimize them with the teacher from scratch, which involves knowledge transfer from teacher to students, as well as interactions between students via mutual learning. Additionally, we develop carefully-

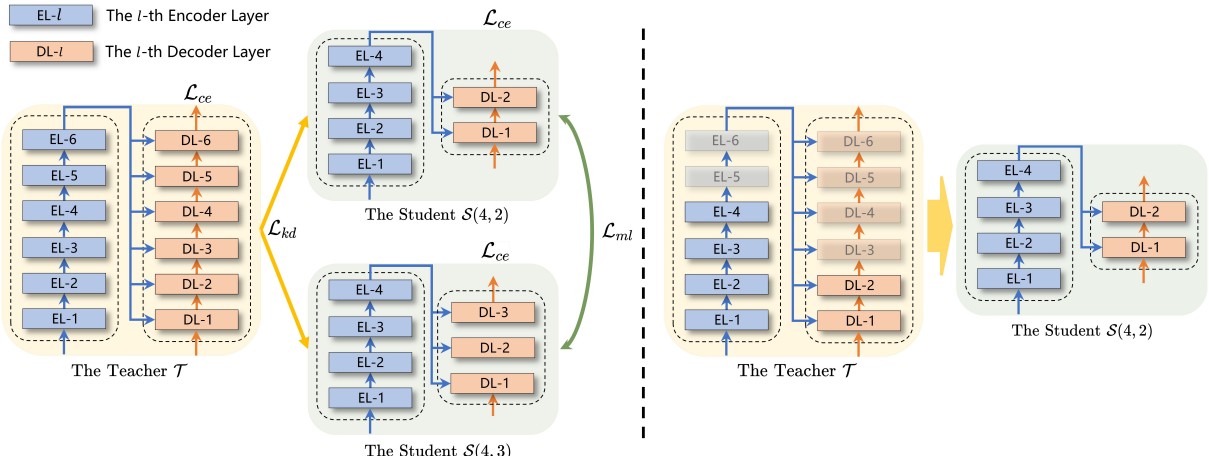

Figure 1: **Left**: An illustration of AIO-KD with sampling 2 students. **Right**: An example of constructing the student $\mathcal{S}(4,2)$ from the teacher $\mathcal{T}$. Employing AIO-KD, we optimize the sample students only at each training step, yet we can still obtain plenty of satisfactory candidate students. $\mathcal{S}(l_e, l_d)$ refers to the student with $l_e$ encoder layers and $l_d$ decoder layers. Note that the teacher is jointly trained with the sample students from scratch, with the students learning the knowledge from the teacher and interacting with other students via mutual learning. $\mathcal{L}_{ce}$, $\mathcal{L}_{kd}$, and $\mathcal{L}_{ml}$ denote the cross-entropy loss, the knowledge distillation loss, and the mutual learning loss, respectively.

designed strategies for AIO-KD, which have been proven to be effective in subsequent experiments.

## 3 All-In-One Knowledge Distillation

In this section, we first give a brief overview of AIO-KD (Section 3.1), and then describe the training objective (Section 3.2). Finally, we detail carefully-designed strategies (Section 3.3).

### 3.1 The Overview of AIO-KD

The left half of Figure 1 provides an overview of AIO-KD. In our work, we adopt the standard Transformer with $\mathcal{N}$ encoder/decoder layers as the teacher $\mathcal{T}$. Inspired by recent studies (Wang et al., 2019; Kasai et al., 2021), we extract fewer-layer subnetworks with deep encoder and shallow decoder from the teacher as the candidate students, which achieve satisfactory performance while maintaining efficient inference. Accordingly, all candidate students can be formalized as $\mathcal{C} = \{\mathcal{S}(l_e, l_d) \mid 1 < l_d \leq l_e \leq \mathcal{N}\}$[1], where $\mathcal{S}(l_e, l_d)$ refers to the student with $l_e$ encoder layers and $l_d$ decoder layers. The right half of Figure 1 gives an example of constructing the student $\mathcal{S}(4,2)$, we extract the adjacent encoder and decoder layers of the teacher, starting from the first layer, to construct it, which shares not only architecture but also parameters with the teacher.

During training, we first randomly and uniformly sample from $\mathcal{C}$ to obatin $\mathcal{K}$ sample students $\{\mathcal{S}_k\}_{k=1}^{\mathcal{K}}$ at each training step.[2] Afterward, we jointly train the teacher and these students, where the students simultaneously learn the teacher's knowledge and interact with other students via mutual learning. Notice that during this process, we carefully develop strategies to accommodate AIO-KD, as described in Section 3.3. When utilized, we re-extract $|\mathcal{C}|$ students from the teacher, satisfying the specifications of various devices.[3]

### 3.2 Training Objective

Overall, the training objective of AIO-KD consists of the following three parts:

$$\mathcal{L} = \mathcal{L}_{ce} + \alpha \mathcal{L}_{kd} + \beta \mathcal{L}_{ml}, \qquad (1)$$

where $\mathcal{L}_{ce}$, $\mathcal{L}_{kd}$, and $\mathcal{L}_{ml}$ denote the cross-entropy loss, the knowledge distillation loss, and the mutual learning loss, $\alpha$ and $\beta$ are two coefficients balancing the effects of different losses, respectively.

**Cross-Entropy Loss $\mathcal{L}_{ce}$** As reported by previous works (Zhang et al., 2018; Guo et al., 2020b), jointly training the teacher and student achieves better knowledge transfer. In this work, the students are optimized jointly with the teacher from scratch. Formally, we decompose $\mathcal{L}_{ce}$ into two parts as follows:

$$\mathcal{L}_{ce} = \mathcal{L}_{ce}^{\mathcal{T}} + \sum_{k=1}^{\mathcal{K}} \mathcal{L}_{ce}^{\mathcal{S}_k}, \qquad (2)$$

---

[1] We follow previous studies (Sun et al., 2021; Ge et al., 2022) to exclude the models with 1-layer decoder because they do not consistently perform well.

[2] In our early exploration, we attempt to explore intelligent sampling strategies for students. However, they do not perform well for AIO-KD.

[3] $|\mathcal{C}| = 1 + 2 + ... + (\mathcal{N} - 1) = \frac{\mathcal{N}(\mathcal{N}-1)}{2}$

where $\mathcal{L}_{ce}^{\mathcal{T}}$ and $\mathcal{L}_{ce}^{\mathcal{S}_k}$ represent the cross-entropy losses for the teacher $\mathcal{T}$ and the student $\mathcal{S}_k$, respectively.

**Knowledge Distillation Loss $\mathcal{L}_{kd}$** Employing AIO-KD, we aim to transfer the teacher's knowledge to multiple students by aligning the students' output probability distributions with those of the teacher, and $\mathcal{L}_{kd}$ is formulated as follows:

$$\mathcal{L}_{kd} = \frac{1}{\mathcal{K}} \sum_{k=1}^{\mathcal{K}} \mathrm{KL}(\mathcal{P}^{\mathcal{T}} || \mathcal{P}^{\mathcal{S}_k}), \qquad (3)$$

where $\mathrm{KL}(\cdot)$ is the Kullback–Leibler distance function, $\mathcal{P}^{\mathcal{T}}$ and $\mathcal{P}^{\mathcal{S}_k}$ denote the output probability distributions of the teacher $\mathcal{T}$ and the student $\mathcal{S}_k$, respectively.

**Mutual Learning Loss $\mathcal{L}_{ml}$** To further promote the students, we incorporate mutual learning to facilitate their interactions, with the loss $\mathcal{L}_{ml}$ defined as

$$\mathcal{L}_{ml} = \frac{2}{\mathcal{K}(\mathcal{K}-1)} \sum_{1 \leq k,k' \leq \mathcal{K}} \mathrm{ML}(\mathcal{P}^{\mathcal{S}_k}, \mathcal{P}^{\mathcal{S}_{k'}}), \quad (4)$$

$$\mathrm{ML}(\mathcal{P}^{\mathrm{S}_k}, \mathcal{P}^{\mathrm{S}_{k'}}) = \begin{cases} \mathrm{KL}(\mathcal{P}^{\mathcal{S}_k} || \mathcal{P}^{\mathcal{S}_{k'}}), & \mathcal{L}_{ce}^{\mathcal{S}_{k'}} \geq \mathcal{L}_{ce}^{\mathcal{S}_k}, \\ \mathrm{KL}(\mathcal{P}^{\mathcal{S}_{k'}} || \mathcal{P}^{\mathcal{S}_k}), & \text{otherwise.} \end{cases}$$
$$(5)$$

Notice that for any two students, the one with the lower cross-entropy loss acts as the senior student, leading the process of mutual learning. As discussed by Liao et al. (2020), such mutual learning is beneficial for NMT.

### 3.3 Carefully-Designed Strategies

**Dynamic Gradient Detaching.** As mentioned in Section 3.2, the teacher and students mutually influence each other through the KD loss. When the performance of the student $\mathcal{S}_k$ is much inferior to that of the teacher $\mathcal{T}$, the gradients $g = \frac{\partial \mathrm{KL}(\mathcal{P}^{\mathcal{T}} || \mathcal{P}^{\mathcal{S}_k})}{\partial \theta_{\mathcal{T}}}$ of the $\mathcal{S}_k$-related KD loss will harm the teacher and the latter may further negatively impact other students, where $\theta_{\mathcal{T}}$ represents the teacher's parameters.

To deal with this issue, we calculate the cross-entropy ratio $\mathcal{L}_{ce}^{\mathcal{S}_k} / \mathcal{L}_{ce}^{\mathcal{T}}$ of the student $\mathcal{S}_k$ to the teacher $\mathcal{T}$, which measures their performance gap. If this ratio exceeds a pre-defined threshold $\eta$, we argue that there exists a significant performance gap between $\mathcal{S}_k$ and $\mathcal{T}$, and thus we do not utilize the gradients $g$ to update the teacher's parameters. To this end, we reformulate the gradients $g$ as fol-

lows:

$$g = \begin{cases} \dfrac{\partial \mathrm{KL}(\mathcal{P}_{\mathcal{T}} || \mathcal{P}_{\mathcal{S}_k})}{\partial \theta_{\mathcal{T}}}, & \dfrac{\mathcal{L}_{ce}^{\mathcal{S}_k}}{\mathcal{L}_{ce}^{\mathcal{T}}} \leq \eta, \\ 0, & \text{otherwise.} \end{cases} \quad (6)$$

**Two-Stage Mutual Learning.** It is worth noting that the students at the early-stage training are often poorly-performed ones, which hinder student interactions. To better leverage the potential of mutual learning, we adopt a two-stage training strategy. At the first training stage, we only utilize the signals from the teacher to guide the training of the students, formulating the training objective as

$$\mathcal{L}_1 = \mathcal{L}_{ce} + \alpha \mathcal{L}_{kd}. \qquad (7)$$

Thereafter, we further introduce $\mathcal{L}_{ml}$ to optimize the students at the second training stage, as shown below:

$$\mathcal{L}_2 = \mathcal{L}_{ce} + \alpha \mathcal{L}_{kd} + \beta \mathcal{L}_{ml}. \qquad (8)$$

## 4 Experiments

### 4.1 Setup

**Datasets.** We conduct experiments on German-English (De-En), English-Romanian (En-Ro), and English-German (En-De) translation tasks. For the De-En task, we use the IWSLT14 De-En corpus, where the training set comprises $160k$ sentence pairs extracted from TED talks. We use the combination of `dev2010` and `dev2012` as the validation set, and the combination of `tst2010`, `tst2011`, and `tst2012` as the test set, respectively. For the En-Ro task, we use the dataset of the WMT16 En-Ro as the training set, containing $610k$ sentence pairs. We separately choose `newsdev2016` and `newstest2016` as the validation and test sets. For the En-De task, we use the WMT14 En-De dataset containing $4.5m$ sentence pairs for training, and we choose `newstest2013` and `newstest2014` as the validation and test sets, respectively. We employ Byte Pair Encoding (BPE) (Sennrich et al., 2016) to split words into subwords. Following common practices, we set the numbers of merging operations as $10k$, $32k$, and $32k$ for the three tasks, respectively. Finally, we report case-sensitive tokenized BLEU (Papineni et al., 2002) as well as COMET (Rei et al., 2020).

**Model Configuration.** We develop AIO-KD and other baselines with `fairseq` (Ott et al., 2019). The standard Transformer with 6 encoder/decoder layers is adopted as the teacher. We use the `transformer_iwslt_de_en` setting for the De-En

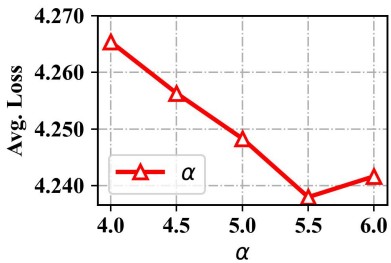 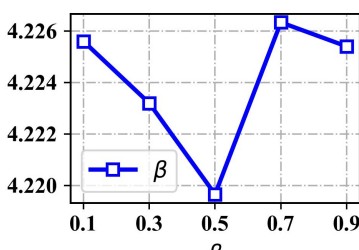 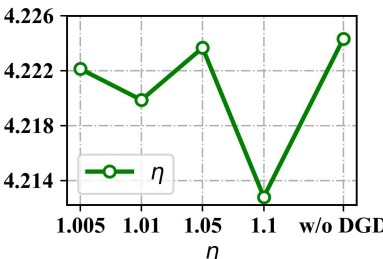

Figure 2: Effects of $\alpha$, $\beta$, and $\eta$ on the validation set of the En-Ro task. "Avg. Loss" refers to the average cross-entropy losses of all candidate students on the validation set, as defined in Equation 9. "w/o DGD" means that we remove *dynamic gradient detaching* from AIO-KD, with no students being detached.

task, and the `transformer_wmt_en_de` setting for the En-Ro and En-De tasks, respectively.

To optimize models, we use Adam (Kingma and Ba, 2015) optimizer with $\beta_1$=0.9, $\beta_2$=0.98, and $\epsilon$=$10^{-9}$. All experiments are conducted on NVIDIA A100 GPUs with mixed-precision training, where the batch sizes are individually set to $4k$, $4k$, and $32k$ tokens for the three tasks. We set the training steps to $300k$, $300k$, and $400k$ per stage for the De-En, En-Ro, and En-De tasks. For other KD baselines, we set the training steps to $200k$ for each student, which are much longer than the average training steps for each student in AIO-KD. We set the number $\mathcal{K}$ of sample student as 2. The selections of hyper-parameters $\alpha$, $\beta$, and $\eta$ are discussed in Section 4.2.

**Model Selection.** In our work, we expect that all candidate students achieve satisfactory performance. However, it is impractical for each student to conduct beam search decoding on the validation set for model selection.

As an efficient alternative, we select the model according to the average cross-entropy losses of all candidate students on the validation set, which can be formulated as

$$\theta^* = \arg\min_{\theta} \frac{1}{|\mathcal{C}|} \sum_{k=1}^{|\mathcal{C}|} \mathcal{L}_{ce}^{\mathcal{S}_k}, \qquad (9)$$

where $\theta^*$ denotes the optimal parameters, which are essentially the teacher's parameters because all candidate students share them.

**Baselines.** We compare our model with the following baselines:

- **Transformer** (Vaswani et al., 2017). It is the most dominant NMT model.
- **Word-KD** (Kim and Rush, 2016). Under Word-KD, the student is optimized to mimic the output probability distributions of the teacher.
- **Selective-KD** (Wang et al., 2021). By using Selective-KD, the student is optimized to mimic

the output probability distributions of the teacher on the complicated words, which have higher cross-entropy losses estimated by the student.

## 4.2 Effects of Hyper-Parameters

We first investigate the effects of $\alpha$, $\beta$, and $\eta$, where $\alpha$ and $\beta$ are used to balance $\mathcal{L}_{kd}$ and $\mathcal{L}_{ml}$ (See Equations 7 and 8), and $\eta$ controls which students' gradients are not utilized to update the teacher's parameters during knowledge transfer (See Section 3.3).

Through our preliminary empirical studies, we find that changes in $\eta$ have negligible impacts on the selection of $\alpha$ and $\beta$. Therefore, we tune $\alpha$ and $\beta$ without *dynamic gradient detaching*, where $\eta$ is not involved. Concretely, we tune $\alpha$ at the first stage and then tune $\beta$ at the second stage with $\alpha$ fixed. Finally, we tune $\eta$ after determining the optimal $\alpha$ and $\beta$.

Figure 2 shows the effects of $\alpha$, $\beta$, and $\eta$ on the En-Ro task, where we set $(\alpha, \beta, \eta)$ to $(5.5, 0.5, 1.1)$. Similarly, we apply above procedures to the De-En and En-De tasks, where $(\alpha, \beta, \eta)$ are separately set to $(5.5, 0.5, 1.1)$ and $(4.5, 0.1, 1.01)$.

## 4.3 Main Results

To demonstrate the superiority of AIO-KD, we report translation quality of all candidate students, as well as training costs.

**Translation Quality.** Table 1 presents BLEU scores of all candidate students on the three tasks. We can draw the following conclusions:

First of all, we observe that both Word-KD and Selective-KD achieve remarkable improvements compared with Transformer, echoing the results reported in previous studies (Kim and Rush, 2016; Wang et al., 2021). Second, AIO-KD significantly outperforms other baselines across all tasks, indicating its effectiveness. Furthermore, the teachers in AIO-KD demonstrate impressive performance.

| Model | 2-2 | 3-2 | 4-2 | 5-2 | 6-2 | 3-3 | 4-3 | 5-3 | 6-3 | 4-4 | 5-4 | 6-4 | 5-5 | 6-5 | 6-6 | Avg. |
|---|---|---|---|---|---|---|---|---|---|---|---|---|---|---|---|---|
| | | | | | | | | IWSLT14 De-En | | | | | | | | |
| Transformer | 33.48 | 34.27 | 34.56 | 34.58 | 34.98 | 34.73 | 34.65 | 34.70 | 35.29 | 34.81 | 34.66 | 35.03 | 34.48 | 34.94 | 35.03 | 34.68 |
| Word-KD | **34.93** | 35.36 | 35.68 | 35.76 | 36.03 | 36.05 | 35.94 | 36.01 | 36.24 | 36.20 | 36.05 | 36.15 | 36.14 | 36.27 | 36.49 | 35.95 |
| Selective-KD | 34.69 | 34.98 | 35.32 | 35.30 | 35.82 | 35.33 | 35.46 | 35.77 | 35.72 | 35.56 | 35.76 | 36.18 | 35.61 | 36.21 | 35.76 | 35.56 |
| AIO-KD (Ours) | 34.64 | **35.99** | **36.62** | **36.73** | **36.85** | **36.43** | **37.15** | **37.22** | **37.25** | **37.25** | **37.36** | **37.45** | **37.53** | **37.45** | **37.69** | **36.91‡** |
| | | | | | | | | WMT16 En-Ro | | | | | | | | |
| Transformer | 30.30 | 30.97 | 31.20 | 30.99 | 31.06 | 31.31 | 30.86 | 31.77 | 31.71 | 31.49 | 31.92 | 31.31 | 31.41 | 31.72 | 32.01 | 31.34 |
| Word-KD | 33.48 | 34.23 | 33.67 | 33.42 | 34.07 | 33.96 | 34.10 | 33.94 | 34.22 | 34.07 | 34.31 | 34.69 | 34.30 | 34.25 | 34.36 | 34.07 |
| Selective-KD | 33.00 | 32.51 | 32.51 | 32.48 | 32.73 | 33.18 | 33.18 | 33.00 | 32.81 | 32.72 | 32.74 | 32.85 | 32.42 | 32.68 | 32.59 | 32.76 |
| AIO-KD (Ours) | **33.97** | **34.50** | **34.84** | **34.80** | **34.77** | **34.87** | **35.17** | **35.15** | **35.11** | **35.29** | **35.40** | **35.32** | **35.39** | **35.47** | **35.44** | **35.03‡** |
| | | | | | | | | WMT14 En-De | | | | | | | | |
| Transformer | 26.08 | 26.25 | 27.05 | 27.56 | 27.48 | 26.14 | 27.12 | 27.41 | 27.66 | 27.39 | 27.50 | 27.94 | 27.88 | 28.17 | 27.98 | 27.31 |
| Word-KD | 26.01 | 26.63 | 27.20 | 27.58 | 27.64 | 26.91 | 27.66 | 27.85 | 27.94 | 27.43 | 27.80 | 27.72 | 28.13 | 28.01 | 28.13 | 27.51 |
| Selective-KD | **26.44** | 26.95 | 27.38 | 27.65 | 27.79 | 27.29 | 27.75 | 28.07 | 28.31 | 27.39 | 28.04 | 28.45 | 28.51 | 28.49 | 28.20 | 27.78 |
| AIO-KD (Ours) | 25.80 | **27.48** | **28.16** | **28.30** | **28.57** | **27.65** | **28.45** | **28.75** | **28.79** | **28.68** | **28.86** | **29.23** | **28.96** | **29.16** | **29.18** | **28.40‡** |

Table 1: Comparisons of BLEU (%) scores of all candidate students. "$l_e$-$l_d$" refers to the candidate student with $l_e$ encoder and $l_d$ decoder layers. "Avg." refers to the average BLEU (%) score of all candidate students. The best results are highlighted in **bold**. We combine the translations from all candidate students for significance test (Koehn, 2004), where "‡" means the improvements over Word-KD and Selective-KD are statistically significant with $p<0.01$.

| Model | 2-2 | 3-2 | 4-2 | 5-2 | 6-2 | 3-3 | 4-3 | 5-3 | 6-3 | 4-4 | 5-4 | 6-4 | 5-5 | 6-5 | 6-6 | Avg. |
|---|---|---|---|---|---|---|---|---|---|---|---|---|---|---|---|---|
| | | | | | | | | IWSLT14 De-En | | | | | | | | |
| Transformer | 33.38 | 35.94 | 36.52 | 37.65 | 38.45 | 37.69 | 37.89 | 38.69 | 39.75 | 38.72 | 38.16 | 39.14 | 38.60 | 39.50 | 40.30 | 38.03 |
| Word-kd | **39.23** | 40.15 | 41.66 | 41.89 | 42.67 | 42.38 | 42.30 | 43.12 | 43.99 | 42.78 | 43.12 | 43.45 | 43.34 | 43.79 | 43.73 | 42.51 |
| Selective-KD | 38.90 | 40.15 | 41.25 | 41.35 | 42.75 | 41.11 | 41.70 | 42.69 | 43.02 | 41.52 | 42.34 | 43.24 | 42.98 | 43.80 | 42.54 | 41.96 |
| AIO-KD | 37.64 | **41.58** | **43.72** | **44.54** | **44.51** | **43.31** | **45.33** | **45.81** | **46.11** | **45.96** | **46.37** | **46.68** | **46.67** | **46.69** | **46.85** | **44.78‡** |
| | | | | | | | | WMT16 En-Ro | | | | | | | | |
| Transformer | 32.12 | 34.12 | 33.87 | 33.54 | 34.32 | 35.41 | 33.61 | 37.35 | 37.62 | 36.92 | 38.02 | 36.56 | 37.32 | 38.41 | 40.54 | 35.98 |
| Word-kd | 46.84 | 48.01 | 47.20 | 47.21 | 48.99 | 48.42 | 50.90 | 50.11 | 50.40 | 50.14 | 51.67 | 52.83 | 50.17 | 50.92 | 50.95 | 49.65 |
| Selective-KD | 41.50 | 40.68 | 42.49 | 42.29 | 41.51 | 41.60 | 42.40 | 43.46 | 45.54 | 45.43 | 44.18 | 43.82 | 43.21 | 43.49 | 44.78 | 43.09 |
| AIO-KD | **47.92** | **51.74** | **52.83** | **52.67** | **53.12** | **52.47** | **53.99** | **54.34** | **54.23** | **54.69** | **54.87** | **54.94** | **55.08** | **55.17** | **55.31** | **53.56‡** |
| | | | | | | | | WMT14 En-De | | | | | | | | |
| Transformer | 36.30 | 39.46 | 41.10 | 42.75 | 43.47 | 41.10 | 42.24 | 44.27 | 44.71 | 43.82 | 44.87 | 45.79 | 45.36 | 46.97 | 46.45 | 43.24 |
| Word-kd | 38.68 | 42.13 | 43.80 | 45.02 | 45.21 | 43.91 | 45.02 | 45.75 | 45.82 | 46.42 | 46.72 | 46.72 | 45.65 | 47.45 | 46.15 | 44.96 |
| Selective-KD | **40.36** | **43.94** | 43.69 | 46.13 | 46.13 | 45.36 | 46.53 | 47.89 | 47.71 | 46.83 | 47.62 | 48.16 | 48.69 | 48.70 | 48.44 | 46.41 |
| AIO-KD | 37.20 | 43.37 | **45.58** | **46.61** | **46.83** | **45.32** | **47.86** | **48.88** | **47.93** | **47.93** | **48.75** | **48.95** | **48.97** | **49.08** | **49.46** | **46.85‡** |

Table 2: Comparisons of COMET (%) scores of all candidate students.

Specifically, the teachers achieve 37.69, 35.44, and 29.18 BLEU scores on the De-En, En-Ro, and En-De tasks, respectively, with improvements of +2.66, +3.43, and +1.20 BLEU scores over Transformer. We attribute the promising improvements of the teachers to the interactions with the students, which will be explored in Section 5.3.

Also, we report the results using COMET metric in Table 2, which support the above conclusions.

**Training Costs.** Apart from the satisfactory performance, the advantages of AIO-KD also owe to its training efficiency. To support our claim, we report the training time and memory usage of each approach, as displayed in Table 3.

First of all, we observe that although adopting both Word-KD and Selective-KD significantly improves model performance, it also brings enormous training costs.

By contrast, AIO-KD is much more eco-friendly. Concretely, GPU hours of AIO-KD spent on training are comparable to those of Transformer on the De-En (29.83 vs. 26.11) and En-Ro (34.83 vs. 33.06) tasks, and much less than those of Word-KD (218.67 vs. 456.67) and Selective-KD (218.67 vs. 406.67) on the En-De task. More encouragingly, AIO-KD also demonstrates its memory-friendliness compared with Transformer on the De-En (16.70 vs. 74.72), En-Ro (55.85 vs. 169.66), and En-De (123.67 vs. 221.22) tasks. Ultimately, AIO-KD saves only one model, i.e., the teacher, highlighting its storage-efficient.

## 4.4 Ablation Studies

To better investigate the effectiveness of the carefully-designed strategies, we compare AIO-KD with the following variants shown in Table 4:

1) **w/o DGD.** In this variant, we discard the

| Model | Training time (GPU hours) / Memory usage (GB) | | |
|---|---|---|---|
| | **IWSLT14 De-En** | **WMT16 En-Ro** | **WMT14 En-De** |
| **Transformer** | 26.11 / 74.72 | 33.06 / 169.66 | 114.44 / 221.22 |
| **Word-KD** | 72.22 / 75.77 | 86.11 / 159.31 | 456.67 / 468.87 |
| **Selective-KD** | 67.22 / 80.65 | 81.67 / 250.86 | 406.67 / 493.33 |
| **AIO-KD (Ours)** | **29.83 / 16.70** | **34.83 / 55.85** | **218.67 / 123.67** |

Table 3: Comparisons of training time (GPU hours) and memory usage (GB). We sum up the training costs of all students for Transformer, Word-KD, and Selective-KD. The presented results are recorded on NVIDIA A100 GPUs.

| Model | 2-2 | 3-2 | 4-2 | 5-2 | 6-2 | 3-3 | 4-3 | 5-3 | 6-3 | 4-4 | 5-4 | 6-4 | 5-5 | 6-5 | 6-6 | Avg. |
|---|---|---|---|---|---|---|---|---|---|---|---|---|---|---|---|---|
| **AIO-KD** | 25.80 | 27.48 | 28.16 | 28.30 | 28.57 | 27.65 | 28.45 | 28.75 | 28.79 | 28.68 | 28.86 | 29.23 | 28.96 | 29.16 | 29.18 | 28.40 |
| w/o DGD | 24.96 | 26.70 | 27.56 | 27.81 | 27.70 | 27.11 | 27.82 | 27.91 | 28.04 | 27.86 | 28.17 | 28.17 | 28.21 | 28.33 | 28.25 | 27.64 |
| w/o ML | 25.60 | 27.32 | 27.92 | 28.20 | 28.45 | 27.59 | 28.26 | 28.45 | 28.54 | 28.55 | 28.77 | 28.97 | 28.74 | 29.10 | 29.08 | 28.24 |
| w/o TST | 25.63 | 27.23 | 28.04 | 28.16 | 28.47 | 27.49 | 28.43 | 28.49 | 28.68 | 28.40 | 28.54 | 28.80 | 28.59 | 28.70 | 28.87 | 28.17 |

Table 4: Ablation studies on the En-De task. "**DGD**" denotes *dynamic gradient detaching*, "**ML**" denotes *mutual learning*, and "**TST**" denotes *two-stage training*.

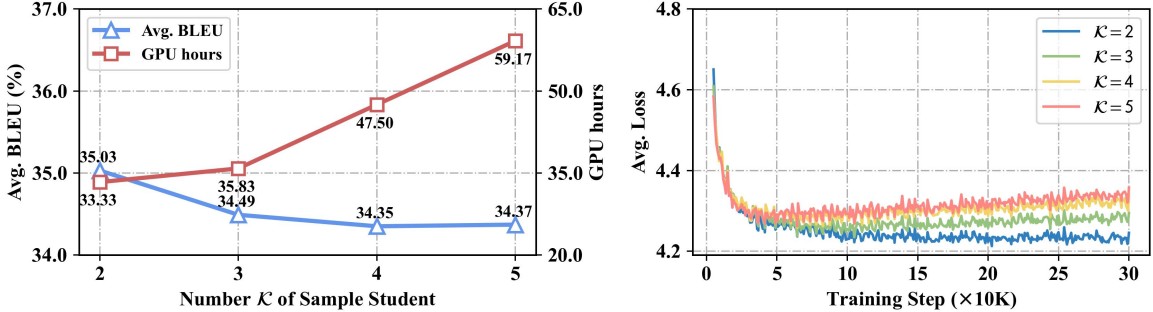

Figure 3: **Left**: Effects of sample student number $\mathcal{K}$ on two metrics: the average BLEU (%) scores of all candidate students on the test set, and GPU hours spent on training. **Right**: Loss curves of AIO-KD on the validation set during training. The results on the En-Ro task are reported.

**D**ynamic **G**radient **D**etaching strategy. It can be found that this removal leads to a significant degeneration of the teacher, with BLEU score dropping from 29.18 to 28.25. Moreover, other candidate students suffer from varying degrees of performance degradation. These results support our claim that poorly-performed students harm the teacher, which further negatively affects other students.

2) **w/o ML.** To verify the benefits of interactions between students, we remove the **M**utual **L**earning loss $\mathcal{L}_{ml}$ from Equation 8. The results show that BLEU scores of all candidate students decrease, indicating that mutual learning indeed further promotes students.

3) **w/o TST.** We employ a one-stage training strategy to train this variant, with the same total training steps as the original AIO-KD. The loss function of this variant is $\mathcal{L}_2$ defined in Equation 8. As expected, AIO-KD benefits from the **T**wo-**S**tage **T**raining strategy across all candidate students, indicating that poorly-performed students at the early-stage training have negative impacts on mutual learning.

## 5  Analysis

### 5.1  Effect of Sample Student Number

In previous experiments, we set the number $\mathcal{K}$ of sample students as 2. A question arises naturally: does increasing $\mathcal{K}$ further improve the students? To answer this question, we experiment with $\mathcal{K}$ ranging from 2 to 5, as illustrated in Figure 3.

In the left half of Figure 3, with an increase of $\mathcal{K}$, the training time of AIO-KD grows from 33.33 to 59.17 GPU hours but does not lead to any performance improvements. Instead, the students degenerate. The right half of Figure 3 also displays loss curves on the validation set with different $\mathcal{K}$, showing that increasing $\mathcal{K}$ leads to the over-fitting problem of the students.

Regarding the above phenomenon, we attribute it to the *gradient conflict problem* (Yu et al., 2020; Liu et al., 2021a; Chai et al., 2022; Yue et al., 2023). Since different students share the parameters of the teacher, when $\mathcal{K}$ increases, the conflict of their gradients becomes more severe during training, ultimately leading to the decline in performance.

| Model | 2-2 | 3-2 | 4-2 | 5-2 | 6-2 | 3-3 | 4-3 | 5-3 | 6-3 | 4-4 | 5-4 | 6-4 | 5-5 | 6-5 | 6-6 | Avg. |
|---|---|---|---|---|---|---|---|---|---|---|---|---|---|---|---|---|
| AIO-KD | 33.97 | 34.50 | 34.84 | 34.80 | 34.77 | 34.87 | 35.17 | 35.15 | 35.11 | 35.29 | 35.40 | 35.32 | 35.39 | 35.47 | 35.44 | 35.03 |
| Seq-KD | 32.30 (-1.67) | 32.46 (-2.04) | 32.52 (-2.32) | 32.69 (-2.11) | 32.89 (-1.88) | 32.61 (-2.26) | 32.69 (-2.48) | 33.28 (-1.87) | 33.13 (-1.98) | 33.11 (-2.18) | 32.94 (-2.46) | 33.10 (-2.22) | 33.08 (-2.31) | 33.25 (-2.22) | 33.41 (-2.03) | 32.90 (-2.13) |
| AIO-KD +Seq-KD | 34.34 (+0.37) | 34.99 (+0.49) | 34.95 (+0.11) | 35.12 (+0.32) | 35.10 (+0.33) | 35.14 (+0.27) | 35.38 (+0.21) | 35.42 (+0.27) | 35.33 (+0.22) | 35.40 (+0.11) | 35.60 (+0.20) | 35.55 (+0.23) | 35.55 (+0.16) | 35.55 (+0.08) | 35.56 (+0.12) | 35.27 (+0.24) |

Table 5: BLEU (%) scores of AIO-KD, Seq-KD, and AIO-KD+Seq-KD on the En-Ro task. "AIO-KD+Seq-KD" means that AIO-KD is conducted on the data provided by Seq-KD. The values in parentheses denote the performance gaps compared with AIO-KD.

| Model | IWSLT14 De-En | WMT16 En-Ro | WMT14 En-De |
|---|---|---|---|
| **Transformer** | 35.01 | 32.01 | 27.98 |
| **SeqMix**† (Guo et al., 2020a) | 36.20 | – | 28.10 |
| **CutOff**† (Shen et al., 2020) | 37.60 | – | 29.10 |
| **PD-R**† (Guo et al., 2022) | – | 34.93 | – |
| **AdMix**† (Jin et al., 2022) | 37.10 | – | 28.26 |
| **CipherDAug**† (Kambhatla et al., 2022) | 37.60 | – | 27.90 |
| **AIO-KD (Ours)** | **37.69** | **35.44** | **29.18** |

Table 6: BLEU (%) scores of the teachers in AIO-KD. "†" means that the results corresponding to the method are taken from the original papers.

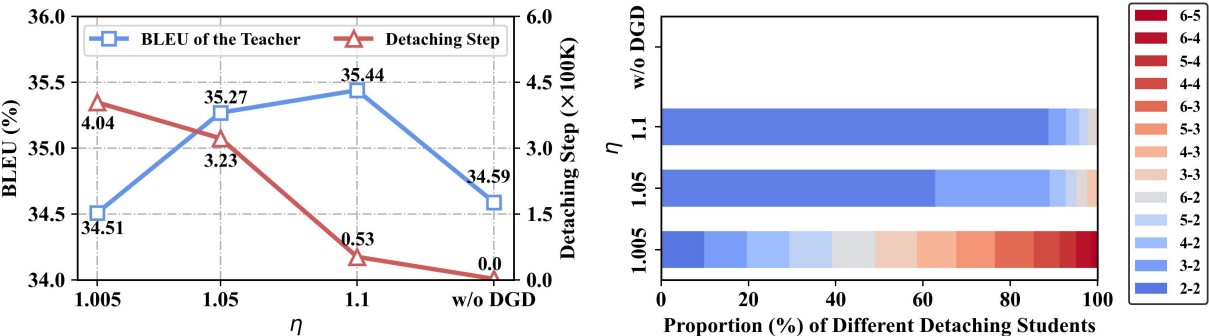

Figure 4: **Left**: Effects of $\eta$ on two metrics: the number of *detaching steps*, and BLEU (%) score of the teacher. **Right**: Proportion (%) of different *detaching students* at *detaching steps*. *Dynamic gradient detaching* varies $\eta$ to protect the teacher from some poorly-performed students at the training steps, where these training steps and students are referred to as *detaching steps* and *detaching students*, respectively. "w/o DGD" means that we remove *dynamic gradient detaching* from AIO-KD, with no students being detached.

## 5.2 Compatibility of AIO-KD with Seq-KD

Seq-KD (Kim and Rush, 2016) is also a widely-used KD approach, where the student is trained on the teacher-generated data. Hence, we explore whether AIO-KD and Seq-KD complement to each other. The results are shown in Table 5.

It is evident that AIO-KD performs significantly better than Seq-KD. Moreover, when AIO-KD and Seq-KD are combined, i.e., AIO-KD+Seq-KD, it achieves an average BLEU score of 35.27, surpassing both AIO-KD (35.27 vs. 35.03) and Seq-KD (35.27 vs. 32.90). These results confirm that AIO-KD and Seq-KD are compatible with each other.

## 5.3 Win-Win Knowledge Distillation

As discussed in Section 4.3, AIO-KD enhances both the teacher and students, making it a win-win KD technique. As shown in Table 6, we compare the enhanced teacher with recently-proposed works on NMT and observe that our model outperforms these strong baselines.

Furthermore, we explore the enhanced teacher from the perspective of model interaction. Under AIO-KD, the teacher is optimized not only to align with labels but also to interact with the students via knowledge transfer. Therefore, we speculate that the teacher's improvements come from these interactions.

To gain deeper insights, we delve into the effects of $\eta$ in *dynamic gradient detaching*, as illustrated in Figure 4. By adjusting $\eta$, the gradients of the KD loss specific to some students at the training steps are not utilized to update the teacher's parameters, where we refer to these training steps and students as *detaching steps* and *detaching students*, respectively.

In the left half of Figure 4, we observe that when $\eta$ decreases, the number of *detaching steps* gradually increases. During this process, the teacher's performance experiences an initial improvement, however, it subsequently undergoes a decline. The above observations reveal that the significant impacts on the teacher through the KD loss.

In the right half of Figure 4, we further present the proportion of different *detaching students* at *detaching steps* corresponding to different $\eta$. We find that when $\eta$=1.1 and $\eta$=1.05, most of poorly-performed students are detached, thus positively impacting the teacher, which severally achieves 35.44 and 35.27 BLEU scores. Conversely, when we set $\eta$ to 1.005, more well-performed students are detached, resulting in a negative impact on the teacher, which achieves 34.51 BLEU score. The above results validate that the teacher benefits from interacting with well-performed students while suffering from the interactions with poorly-performed ones.

Overall, our analyses suggest that the teacher can derive benefits from the weaker students, offering valuable insights for future research.

## 6 Conclusion

In this work, we present AIO-KD, a novel KD framework for NMT that constructs various candidate students from the teacher itself. With AIO-KD, we jointly optimize the teacher and the sample students from scratch. During this process, the students learn from the teacher and interact with other students via mutual learning, resulting in multiple satisfactory students with efficient training. Carefully-designed strategies are also introduced to accommodate AIO-KD. Extensive experiments and in-depth analyses on three benchmarks demonstrate the superiority of our AIO-KD.

In the future, we plan to explore more compact subnetworks of teacher as students using parameter pruning methods. Additionally, we aim to extend AIO-KD to large language models (LLMs) to validate its generalizability.

## Limitations

As mentioned above, the students in AIO-KD are derived from the teacher and they share parameters. Such a design yields multiple high-quality students with significantly reduced training costs, compared with conventional KD approaches. However, its limitation is that the students possess the same model architecture as the teacher. Besides, despite achieving impressive efficiency and performance, our work is only conducted based on Transformer. Thus, we plan to validate AIO-KD on more model architectures in future work.

## Ethics Statement

This work aims to explore an eco-friendly KD approach for NMT, and we hope our method can inspire future work. Our work does not involve any data collection. In the experiments, all the datasets are publicly available and commonly used in the NMT community. Besides, we develop AIO-KD and other baselines based on a widely-used open-source tool fairseq (Ott et al., 2019). The comparisons in this work are conducted based on the same experimental settings and datasets.

## Acknowledgements

The project was supported by National Natural Science Foundation of China (Nos. 62036004, 62276219, 62076211), Natural Science Foundation of Fujian Province of China (No. 2020J06001), and University-Industry Cooperation Programs of Fujian Province of China (No. 2023H6001). We also thank the reviewers for their insightful comments.

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
