# OpenReview forum: "Exploring All-In-One Knowledge Distillation Framework for Neural Machine Translation"
_EMNLP/2023/Conference — EMNLP 2023 Main_

### Official Review · Reviewer_s6c5 · 2023-08-03

**Soundness:** 4

**Excitement:**

3: Ambivalent: It has merits (e.g., it reports state-of-the-art results, the idea is nice), but there are key weaknesses (e.g., it describes incremental work), and it can significantly benefit from another round of revision. However, I won't object to accepting it if my co-reviewers champion it.

**Paper Topic And Main Contributions:**

The authors propose All-in-One Knowledge Distillation (AIO-KD) which (1) randomly extract subnetworks from the teacher as students (2) jointly optimize the teacher and students. They leverage some tricks like stopping training loss from being affected by poorly performing students.

**Questions For The Authors:**

- Am I reading this correctly that increasing the number of students *degrades* performance?

**Reasons To Accept:**

- This paper combines online KD and Mutual Learning and adds some nice tricks that seem to improve empirical results compared to Seq-KD and incorporating both simultaneously brings additional minor gains.
- Ablation studies are sound and appreciated the analysis of varying the number of students.


**Reasons To Reject:**

- Online distillation (where the teacher and student are jointly optimized on some loss Lce) has been extensively explored in the past (see section 5.3 in https://arxiv.org/pdf/2006.05525.pdf)
- Mutual learning (wherein students learn from each-other) has also been explored.
- The authors should have also considered more intelligent ways to sample from the subnetworks of a larger teacher model rather than just randomly sampling encoder and decoder layers.

**Reproducibility:**

4: Could mostly reproduce the results, but there may be some variation because of sample variance or minor variations in their interpretation of the protocol or method.

**Reviewer Confidence:**

4: Quite sure. I tried to check the important points carefully. It's unlikely, though conceivable, that I missed something that should affect my ratings.

---

> ### Author Rebuttal · Authors · 2023-08-28
>
> $\textbf{1. [Reply to Q1 and Q2 in Reasons to Reject]}$ It is worth noting that our contribution lies in proposing an efficient knowledge distillation framework for multi-student scenarios. The innovation of our framework far exceeds the simple combination of knowledge distillation (KD) and mutual learning (ML), as our framework is not limited to any specific KD or ML techniques.
>
> $\textbf{2. [Reply to Q3 in Reasons to Reject]}$ We appreciate your valuable suggestions. In our early exploration, we also attempt to explore intelligent sampling strategies for students. For instance, we sort all the candidate students according to the ratio of  the student's parameter count to that of the teacher, and then gradually sample the candidate students in descending/increasing order of this ratio. After exploring some sampling strategies, we find that they do not consistently perform well for our AIO-KD. Hence, we default to sampling students uniformly at random during training. We will clarify the student sampling strategy in the revision and explore more effective student sampling strategies in future work.
>
> $\textbf{3. [Reply to Q1 in Questions For The Authors]}$ Thank you for the question. Yes, increasing the number of sampled students degrades performance. In Section 5.2, we investigate the impact of the sample student number $\mathcal{K}$, i.e., the number of the students sampled at each training step. In the left half of Figure 3, we vary $\mathcal{K}$ from 2 to 5 and observe that increasing $\mathcal{K}$ does not lead to further performance improvements but instead results in a significant increase in training costs.

---

### Official Review · Reviewer_TCJa · 2023-08-04

**Soundness:** 4

**Excitement:**

3: Ambivalent: It has merits (e.g., it reports state-of-the-art results, the idea is nice), but there are key weaknesses (e.g., it describes incremental work), and it can significantly benefit from another round of revision. However, I won't object to accepting it if my co-reviewers champion it.

**Paper Topic And Main Contributions:**

The paper proposes a distillation framework where a few student models are trained simultaneously with the teacher model in order to obtain a series of smaller models at the end of knowledge distillation. Each of the student models is a sub-part of the teacher model (i.e. they share the exactly same weight parameters for the sub-part and are stored in the same place in memory). In the proposed method AIO-KD, knowledge distillation does not only take place between the teacher model and each student model, but also between the student models via mutual learning.

The authors present a few challenges when the algorithm attempts to train the teacher model and student models simultaneously, such as the potential bad influence of immature student models in the early training stage. The paper then proposes corresponding solutions that dynamically detach gradients and employ mutual learning only during the second stage in a two-stage training scheme.

**Questions For The Authors:**

A. In line 229, why 200k training steps for student models obtained using other distillation algorithms are considered “much longer than the average training steps for each student in AIO-KD”? The student models in AIO-KD are obtained with the same number of training steps as the teacher model. It does not seem to make sense to take the average. Does this different number of training steps make the comparison of GPU hours to evaluate computational costs unfair?

B. What is the student model architecture (for Word-KD and Selective-KD) used to obtain training time and memory usage statistics in Table 2?

**Reasons To Accept:**

1. The engineering part of the paper seems to be well thought out and neatly executed. The empirical results demonstrate the effectiveness of AIO-KD.

2. Most analyses of the proposed algorithm are interesting and insightful. The authors have conducted extensive experiments to understand the influence of a wide range of hyper-parameters or variables in AIO-KD (e.g., the number of student models, and ablation studies for each component in the overall loss function). These analyses are helpful for us to understand the advantages and limitations of the proposed method.

**Reasons To Reject:**

1. The paper has a very strong assumption that a reader is familiar with the task of neural machine translation, and hence, some preliminaries like input representations, model outputs, precise loss functions, etc. are omitted. This raises a barrier for a wider range of audience to understand the paper well.

2. While the paper presents a lot of interesting analyses, some are left without a thorough explanation or a clear conclusion. For example, the trends of some hyper-parameters seem rather random in Sec 4.2. Is this randomness a result of statistical noises alone? How does Fig. 2 teach us a strategy for hyper-parameter selection?

3. A few technical details could be further clarified. Otherwise, some claims may not be well supported. Kindly refer to the "Questions for the Authors" section.

**Reproducibility:**

4: Could mostly reproduce the results, but there may be some variation because of sample variance or minor variations in their interpretation of the protocol or method.

**Reviewer Confidence:**

3: Pretty sure, but there's a chance I missed something. Although I have a good feel for this area in general, I did not carefully check the paper's details, e.g., the math, experimental design, or novelty.

---

> ### Author Rebuttal · Authors · 2023-08-28
>
> $\textbf{1. [Reply to Q1 in Reasons to Reject]}$ Thanks for your kind suggestions.  Due to page limitations, the introduction to neural machine translation is omitted. We will supplement related preliminaries in the later revision.
>
> $\textbf{2. [Reply to Q2 in Reasons to Reject]}$ We will check the paper again in light of your feedback and supplement more conclusions in the analysis. Regarding hyper-parameter selection, it is time-consuming to determine 3 hyper-parameters in AIO-KD using grid search. Instead, we turn to adopting a more efficient heuristic method based on our preliminary empirical results (See Section 4.2).  In fact, our AIO-KD is relatively insensitive to these hyper-parameters. We will supplement the impacts of hyper-parameters of all translation tasks in the later revision to support our claim.
>
> $\textbf{3. [Reply to Q1 in Questions For The Authors]}$ There may be some misunderstandings about our work. At each training step, the teacher is always involved, while only $\mathcal{K}$ students are sampled from |$\mathcal{C
> }$| candidate students for training. This means that each candidate student does not obtain the same number of training steps as the teacher. Since the sampling procedure is random and independent, the expected number of training steps for each candidate student in our AIO-KD can be represented as $\mathcal{T}$ $\times$  $\mathcal{K}$ $/$ |$\mathcal{C
> }$|, where $\mathcal{T}$ refers to the total number of training steps including the first and second stages.  For the De-En and En-Ro tasks, the expected number of training steps per student is (300K+300K) $\times$ 2/15 = 80K steps, which is less than 200K steps, and for the En-De task, it is  (400K+400K) $\times$ 2/15 $\approx$ 107K steps, which is also less than 200K steps. These results support our claim in Section 4.1 that the average number of training steps for each student in our AIO-KD is much shorter than that of the student in other KD baselines.
>
> $\textbf{4. [Reply to Q2 in Questions For The Authors]}$ Following previous work [1,2], both the students and teacher in our work adopt the dominant Transformer architecture, with the only difference being that the students have fewer encoder and decoder layers compared with the teacher.
>
> $\textbf{References}$
>
> [1] Wang et al. Selective Knowledge Distillation for Neural Machine Translation. 2021. In Proc. of ACL.
>
> [2] Liang et al. Multi-Teacher Distillation with Single Model for Neural Machine Translation. 2022. IEEE/ACM Trans. Audio, Speech, Language Process.

---

### Official Review · Reviewer_gH6v · 2023-08-05

**Soundness:** 4

**Excitement:**

3: Ambivalent: It has merits (e.g., it reports state-of-the-art results, the idea is nice), but there are key weaknesses (e.g., it describes incremental work), and it can significantly benefit from another round of revision. However, I won't object to accepting it if my co-reviewers champion it.

**Paper Topic And Main Contributions:**

Conventional knowledge distillation (KD) approaches have to be conducted multiple times to train student models of varying sizes that fit different devices. To address this issue, this paper proposes a framework based on the Transformer to train various student models at the same time and consider the correlation among them as well. To further improve the KD performance, two strategies, dynamic gradient detaching and two-stage mutual learning, are also proposed to mitigate the negative impact caused by poorly-performed students.

**Questions For The Authors:**

[Table 2] Is the training time of Transformer, Word-KD and Selective-KD the sum of the training time of each student (2-2 to 6-6)?

**Reasons To Accept:**

1 . The proposed framework seems sound.
2. The experimental analysis given is sufficient and can support their claims.
3. The paper is well-organized and easy to understand.

**Reasons To Reject:**

The compared baselines, Word-KD, Seq-KD and Selective-KD are not the latest. More recent works should be compared.

**Reproducibility:**

4: Could mostly reproduce the results, but there may be some variation because of sample variance or minor variations in their interpretation of the protocol or method.

**Reviewer Confidence:**

4: Quite sure. I tried to check the important points carefully. It's unlikely, though conceivable, that I missed something that should affect my ratings.

---

> ### Author Rebuttal · Authors · 2023-08-28
>
> $\textbf{1. [Reply to Q1 in Reasons To Reject]}$ We compare our AIO-KD with Word-KD, Seq-KD, and Selective-KD because they are widely used in neural machine translation (NMT). As discussed in our paper, our main contribution is proposing an efficient multi-student KD framework that is compatible with various KD techniques, including recent ones. In future, we will attempt to incorporate more KD techniques into our framework and validate it on various NLP tasks beyond NMT.
>
> $\textbf{2. [Reply to Q1 in Question For The Authors]}$ Thank you for the question. Yes, we sum up the training time of all candidate students as the training time of Transformer, Word-KD, and Selective-KD. Since each baseline obtains one student at a time, we determine its training costs by adding up the training costs of all candidate students. Overall, our AIO-KD is much more efficient, as it only needs to be executed once to obtain multiple high-quality students.  In the revision, we will clarify these points in the caption of Table 2.

---

### Meta-Review · Area_Chair_9AeK · 2023-09-25

**Recommendation:** 3

**Metareview:**

This paper provides a framework for jointly distilling a teacher to multiple student networks. It presents a few challenges eg the potential bad influence of immature student models in the early training stage. The paper then proposes to dynamically detach gradients and employ mutual learning only during the second stage in a two-stage training scheme. The empirical results demonstrate the effectiveness of AIO-KD. The analyses of the proposed algorithm are interesting and insightful.

---

### Decision · Program_Chairs · 2023-10-07

**Decision:**

Accept-Main

**Comment:**

This paper provides a framework for jointly distilling a teacher to multiple student networks. It presents a few challenges eg the potential bad influence of immature student models in the early training stage. The paper then proposes to dynamically detach gradients and employ mutual learning only during the second stage in a two-stage training scheme. The empirical results demonstrate the effectiveness of AIO-KD. The analyses of the proposed algorithm are interesting and insightful.